# Accelerating Reinforcement Learning with Value-Conditional State Entropy Exploration

**Dongyoung Kim**
KAIST

**Jinwoo Shin**
KAIST

**Pieter Abbeel**
UC Berkeley

**Younggyo Seo**[*]
KAIST

## Abstract

A promising technique for exploration is to maximize the entropy of visited state distribution, *i.e.,* state entropy, by encouraging uniform coverage of visited state space. While it has been effective for an unsupervised setup, it tends to struggle in a supervised setup with a task reward, where an agent prefers to visit high-value states to exploit the task reward. Such a preference can cause an imbalance between the distributions of high-value states and low-value states, which biases exploration towards low-value state regions as a result of the state entropy increasing when the distribution becomes more uniform. This issue is exacerbated when high-value states are narrowly distributed within the state space, making it difficult for the agent to complete the tasks. In this paper, we present a novel exploration technique that maximizes the *value-conditional state entropy*, which separately estimates the state entropies that are conditioned on the value estimates of each state, then maximizes their average. By only considering the visited states with similar value estimates for computing the intrinsic bonus, our method prevents the distribution of low-value states from affecting exploration around high-value states, and vice versa. We demonstrate that the proposed alternative to the state entropy baseline significantly accelerates various reinforcement learning algorithms across a variety of tasks within MiniGrid, DeepMind Control Suite, and Meta-World benchmarks. Source code is available at https://sites.google.com/view/rl-vcse.

## 1 Introduction

Recent advances in exploration techniques have enabled us to train strong reinforcement learning (RL) agents with fewer environment interactions. Notable techniques include injecting noise into action or parameter spaces (Sehnke et al., 2010; Rückstiess et al., 2010; Wawrzynski, 2015; Lillicrap et al., 2016; Fortunato et al., 2018) and using visitation counts (Thrun, 1992b; Bellemare et al., 2016; Sutton & Barto, 2018; Burda et al., 2019) or errors from predictive models (Stadie et al., 2015; Pathak et al., 2017, 2019; Sekar et al., 2020) as an intrinsic reward. In particular, a recently developed approach that maximizes the entropy of visited state distribution has emerged as a promising exploration technique (Hazan et al., 2019; Lee et al., 2019; Mutti et al., 2021) for unsupervised RL, where the agent aims to learn useful behaviors without any task reward (Liu & Abbeel, 2021a,b; Yarats et al., 2021b).

The idea to maximize the state entropy has also been utilized in a supervised setup where the task reward is available from environments to improve the sample-efficiency of RL algorithms (Tao et al., 2020; Seo et al., 2021; Nedergaard & Cook, 2022; Yuan et al., 2022). Notably, Seo et al. (2021) have shown that maximizing the sum of task reward and intrinsic reward based on a state entropy estimate can accelerate RL training. However, in this supervised setup, we point out that this approach often suffers from an imbalance between the distributions of high-value and low-value states, which occurs as an agent prefers to visit high-value states for exploiting the task reward. Because state entropy increases when the distribution becomes more uniform, low-value states get to receive a higher

---

[*]Now at Dyson Robot Learning Lab. Correspondence to younggyo.seo@dyson.com.

37th Conference on Neural Information Processing Systems (NeurIPS 2023).

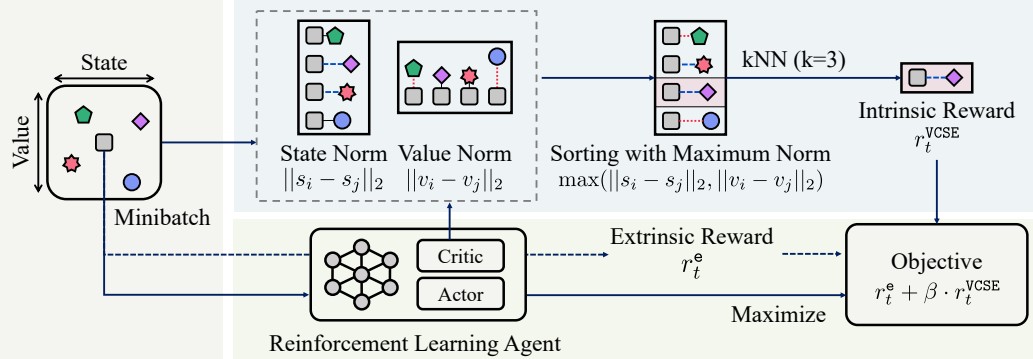

Figure 1: Illustration of our method. We randomly sample states from a replay buffer and compute the Euclidean norm in state and value spaces using pairs of samples within a minibatch. We then sort the samples based on their maximum norms. We find the $k$-th nearest neighbor among samples (*e.g.,* $k = 3$ in the figure) and use the distance to it as an intrinsic reward. Namely, our method excludes the samples whose values significantly differ for computing the intrinsic reward. Then we train our RL agent to maximize the sum of the intrinsic reward and the extrinsic reward.

intrinsic bonus than high-value states, which biases exploration towards low-value states. This makes it difficult for the agent to explore the region around high-value states crucial for solving target tasks, which exacerbates when high-value states are narrowly distributed within the state space.

In this paper, we present a novel exploration technique that maximizes *value-conditional state entropy* which separately estimates the state entropies that are conditioned on the value estimates of each state, then maximizes their average. This intuitively can be seen as partitioning the state space with value estimates and maximizing the average of state entropies of partitioned spaces. Namely, our method avoids the problem of state entropy maximization by preventing the distribution of low-value states from affecting exploration around high-value states, and vice versa, by only considering the states with similar value estimates for computing the intrinsic bonus. For value-conditional state entropy estimation, we utilize the Kraskov-Stögbauer-Grassberger estimator (Kraskov et al., 2004) along with a value normalization scheme that makes value distribution consistent throughout training. We define our intrinsic reward as proportional to the value-conditional state entropy estimate and train RL agents to maximize the sum of task reward and intrinsic reward.

We summarize the main contributions of this paper:

- We present a novel exploration technique that maximizes *value-conditional state entropy* which addresses the issue of state entropy exploration in a supervised setup by taking into account the value estimates of visited states for computing the intrinsic bonus.

- We show that maximum value-conditional state entropy (VCSE) exploration successfully accelerates the training of RL algorithms (Mnih et al., 2016; Yarats et al., 2021a) on a variety of domains, such as MiniGrid (Chevalier-Boisvert et al., 2018), DeepMind Control Suite (Tassa et al., 2020), and Meta-World (Yu et al., 2020) benchmarks.

## 2 Related Work

**Exploration in RL** Exploration has been actively studied to solve sparse reward tasks or avoid being stuck at local optima. One of the classical approaches to encourage exploration is $\epsilon$-greedy algorithm (Sutton & Barto, 2018). This idea has been extended to approaches that inject noise into the action space (Wawrzynski, 2015; Lillicrap et al., 2016) or parameter space (Sehnke et al., 2010; Rückstiess et al., 2010; Fortunato et al., 2018; Plappert et al., 2018) and approaches that maximize the action entropy (Ziebart, 2010; Haarnoja et al., 2018). Similarly to our work that introduces an intrinsic reward, there have been approaches that use the errors from predictive models as an intrinsic reward (Schmidhuber, 1991; Oudeyer et al., 2007; Stadie et al., 2015; Pathak et al., 2017, 2019; Sekar et al., 2020; Badia et al., 2020). Instead of using the knowledge captured in the model, we use a metric that can be quantified from data. The idea of using the state visitation count as an intrinsic

reward (Thrun, 1992b; Bellemare et al., 2016; Tang et al., 2017; Ostrovski et al., 2017; Burda et al., 2019) is also related. However, our method differs in that we directly maximize the diversity of data. Our work also differs from a line of work that balances exploration and exploitation (Thrun, 1992a; Brafman & Tennenholtz, 2002; Tokic, 2010; Whitney et al., 2021; Schäfer et al., 2022) as we instead consider a situation where exploitation biases exploration towards specific state space regions.

**Maximum state entropy exploration**   The approach most related to our work is to utilize state entropy as an intrinsic reward. A popular application of maximizing state entropy is unsupervised RL (Lee et al., 2019; Hazan et al., 2019; Mutti & Restelli, 2020; Mutti et al., 2021; Liu & Abbeel, 2021b; Yarats et al., 2021b; Zhang et al., 2021; Guo et al., 2021; Mutti et al., 2022a,b; Yang & Spaan, 2023), where the agent learns useful behaviors without task reward. We instead consider a setup where task reward is available. A potentially related approach is of Yang & Spaan (2023) which introduces safety constraint for exploration in that one can also consider the approach of introducing a value constraint for exploration. However, our work differs in that our motivation is not to discourage exploration on specific state regions. The work closest to ours is approaches that make agents maximize state entropy along with task rewards (Tao et al., 2020; Seo et al., 2021; Nedergaard & Cook, 2022; Yuan et al., 2022). We point out they often suffer from an imbalance between distributions of high-value and low-value states and propose to take into account the value estimates to address this issue.

## 3   Preliminaries

**Definition**   Let $Z = (X, Y)$ be a random variable whose joint entropy is defined as $\mathcal{H}(X, Y) = -E_{(x,y) \sim p(x,y)}[\log p(x, y)]$ and marginal entropy is defined as $\mathcal{H}(X) = -E_{x \sim p_X(x)}[\log p_X(x)]$. The conditional entropy is defined as $\mathcal{H}(Y \mid X) = E_{(x,y) \sim p(x,y)}[\log p(y \mid x)]$, which quantifies the amount of information required for describing the outcome of Y given that the value of X is known.

**Reinforcement learning**   We formulate a problem as a Markov decision process (MDP; Sutton & Barto 2018), which is defined as a tuple $(\mathcal{S}, \mathcal{A}, p, r^{\text{e}}, \gamma)$. Here, $\mathcal{S}$ denotes the state space, $\mathcal{A}$ denotes the action space, $p(s_{t+1}|s_t, a_t)$ is the transition dynamics, $r^{\text{e}}$ is the extrinsic reward function $r_t^{\text{e}} = r^{\text{e}}(s_t, a_t)$, and $\gamma \in [0, 1)$. Then we train an agent to maximize the expected return.

### 3.1   Maximum State Entropy Exploration

**$k$-nearest neighbor entropy estimator**   When we aim to estimate the entropy of $X$ but the density $p_X$ is not available, non-parametric entropy estimators based on $k$-nearest neighbor distance can be used. Specifically, given $N$ i.i.d. samples $\{x_i\}_{i=1}^N$, the Kozachenko-Leonenko (KL) entropy estimator (Kozachenko & Leonenko, 1987; Singh et al., 2003; Kraskov et al., 2004) is defined as:

$$\widehat{H}_{\text{KL}}(X) = -\psi(k) + \psi(N) + \log c_{d_X} + \frac{d_X}{N} \sum_{i=1}^{N} \log D_x(i), \qquad (1)$$

where $\psi$ is the digamma function, $D_x(i)$ is twice the distance from $x_i$ to its $k$-th nearest neighbor, $d_X$ is the dimensionality of X, and $c_{d_X}$ is the volume of the $d_X$-dimensional unit ball.[2]

**State entropy as an intrinsic reward**   Let $\mathcal{B}$ be a replay buffer and $S$ be a random variable of dimension $d_s$ with a probability density $p_S(s) = \sum_{s \in \mathcal{B}} \mathbf{1}_{S=s}/|\mathcal{B}|$. The main idea of maximum state entropy exploration is to encourage an agent to maximize the state entropy $\mathcal{H}(S)$ (Liu & Abbeel, 2021b; Seo et al., 2021). The KL entropy estimator in Equation 1 is used to estimate the state entropy $\widehat{H}_{\text{KL}}(S)$, and the intrinsic reward $r_i^{\text{SE}}$ is defined by ignoring constant terms and putting logarithm along with an additive constant 1 for numerical stability as below:

$$\widehat{H}_{\text{KL}}(S) = -\psi(k) + \psi(N) + \log c_{d_S} + \frac{d_S}{N} \sum_{i=1}^{N} \log D_s(i) \qquad r_i^{\text{SE}} = \log(D_s(i) + 1) \qquad (2)$$

where $D_s(i)$ is twice the distance from $s_i$ to its $k$-nearest neighbor.

---

[2]For instance, $c_{d_X} = \widehat{\pi}^{d_X/2}/\Gamma(1 + d_X/2)$ for Euclidean norm where $\widehat{\pi} \approx 3.14159$ is a constant.

## 3.2 Conditional Entropy Estimation

**Naive conditional entropy estimator** Given $N$ i.i.d. samples $\{z_i\}_{i=1}^{N}$ where $z_i = (x_i, y_i)$, the conditional entropy can be estimated by using the chain rule of conditional entropy $H(Y \mid X) = H(X, Y) - H(X)$. Specifically, one can compute $\widehat{H}_{\text{KL}}(X, Y)$ and $\widehat{H}_{\text{KL}}(X)$ and use these estimates for computing $\widehat{H}(Y \mid X)$. But a major drawback of this naive approach is that it does not consider length scales in spaces spanned by $X$ and $Y$ can be very different, *i.e.,* the scale of distances between samples and their $k$-nearest neighbors in each space could be very different.

**KSG conditional entropy estimator** To address the issue of estimating entropy in different spaces spanned by $X$ and $Y$, we employ Kraskov-Stögbauer-Grassberger (KSG) estimator (Kraskov et al., 2004), which is originally designed for estimating mutual information. The main idea of KSG estimator is to use different $k$-values in the joint and marginal spaces for adjusting the different length scales across spaces. Given a sample $z_i$ in the joint space and the maximum norm $||z - z'||_{\text{max}} = \max\{||x - x'||, ||y - y'||\}$,[3] we find a $k$-th nearest neighbor $z_i^{k\text{NN}} = (x_i^{k\text{NN}}, y_i^{k\text{NN}})$. To find a value that makes $x_i^{k\text{NN}}$ be $n_x(i)$-th nearest neighbor of $x_i$ in the space spanned by $X$, we count the number $n_x(i)$ of points $x_j$ whose distances to $x_i$ are less than $\epsilon_x(i)/2$, where $\epsilon_x(i)$ is twice the distance from $x_i$ to $x_i^{k\text{NN}}$ (see Appendix D for an illustrative example). We note that $\epsilon_y(i)$ and $n_y(i)$ can be similarly defined by replacing $x$ by $y$. Now let $\epsilon(i)$ be twice the distance between $z_i$ and $z_i^{k\text{NN}}$, *i.e.,* $\epsilon(i) = 2 \cdot ||z_i - z_i^{k\text{NN}}||_{\text{max}}$. Then KSG estimators for the joint and marginal entropies are given as:

$$\widehat{H}_{\text{KSG}}(X, Y) = -\psi(k) + \psi(N) + \log(c_{d_X} c_{d_Y}) + \frac{d_X + d_Y}{N} \sum_{i=1}^{N} \log \epsilon(i) \tag{3}$$

$$\widehat{H}_{\text{KSG}}(X) = -\frac{1}{N} \sum_{i=1}^{N} \psi(n_x(i) + 1) + \psi(N) + \log c_{d_X} + \frac{d_X}{N} \sum_{i=1}^{N} \log \epsilon_x(i) \tag{4}$$

Then we use the chain rule of conditional entropy $\widehat{H}_{\text{KSG}}(X, Y) - \widehat{H}_{\text{KSG}}(X)$ and estimators from Equation 3 and Equation 4 to obtain a conditional entropy estimator and its lower bound as below:

$$\begin{aligned}
\widehat{H}_{\text{KSG}}(Y \mid X) &= \frac{1}{N} \sum_{i=1}^{N} \left[ \psi(n_x(i) + 1) + d_X(\log \epsilon(i) - \log \epsilon_x(i)) + d_Y \log \epsilon(i) \right] + C \\
&\geq \frac{1}{N} \sum_{i=1}^{N} \left[ \psi(n_x(i) + 1) + d_Y \log \epsilon(i) \right] - \psi(k) + \log c_{d_Y}
\end{aligned} \tag{5}$$

where $C$ denotes $-\psi(k) + \log c_{d_Y}$ and lower bounds holds because $\epsilon(i) \geq \epsilon_x(i)$ always holds.

## 4 Method

We present a new exploration technique that maximizes the *value-conditional* state entropy (VCSE), which addresses the issue of state entropy exploration in a supervised setup by taking into account the value estimates of visited states for computing the intrinsic bonus. Our main idea is to prevent the distribution of high-value states from affecting exploration around low-value states, and vice versa, by filtering out states whose value estimates significantly differ from each other for computing the intrinsic bonus. In this section, we first describe how to define the value-conditional state entropy and use it as an intrinsic reward (see Section 4.1). We then describe how we train RL agents with the intrinsic reward (see Section 4.2). We provide the pseudocode of our method in Algorithm 1.

### 4.1 Maximum Value-Conditional State Entropy

**Value-conditional state entropy** Let $\pi$ be a stochastic policy, $f_v^{\pi}$ be an extrinsic critic function, and $V_{\pi}^{\text{vs}}$ be a random variable with a probability density $p_{V_{\pi}^{\text{vs}}}(v) = \sum_{s \in \mathcal{B}} \mathbf{1}_{f_v^{\pi}(s) = v} / |\mathcal{B}|$. Our key idea is to maximize the *value-conditional* state entropy $\mathcal{H}(S \mid V_{\pi}^{\text{vs}}) = E_{v \sim p_{V_{\pi}^{\text{vs}}}(v)}[\mathcal{H}(S \mid V_{\pi}^{\text{vs}} = v)]$, which

---

[3]$z$ and $z'$ are arbitrary samples from a random variable $Z$. We also note that this work uses Euclidean norm for $||x - x'||$ and $||y - y'||$, while any norm can be used for them.

**Algorithm 1** Maximum Value-Conditional State Entropy Exploration

---

1: Initialize policy $\pi$, extrinsic critic $f_v^\pi$, critic $f_{v,\text{T}}^\pi$, replay buffer $\mathcal{B}$
2: **for** each environment step $t$ **do**
3:     Collect transition $\tau_t = (s_t, a_t, s_{t+1}, r_t^{\text{e}})$ and update $\mathcal{B} \leftarrow \mathcal{B} \cup \{\tau_t\}$
4:     Sample mini-batch $\{\tau_{t_j}\}_{j=1}^B \sim \mathcal{B}$
5:     **for** $j = 1$ **to** $B$ **do**
6:         Find $(s_{t_j}^{k\text{NN}}, v_{t_j}^{k\text{NN}})$ from $\{\tau_{t_j}\}_{j=1}^B$ using $\max(||s - s'||, ||v - v'||)$
7:         Compute distances $\epsilon_s(t_j) \leftarrow 2 \cdot ||s_{t_j} - s_{t_j}^{k\text{NN}}||$ and $\epsilon_v(t_j) \leftarrow 2 \cdot ||v_{t_j} - v_{t_j}^{k\text{NN}}||$
8:         Compute joint state-value space distance $\epsilon(t_j) \leftarrow \max(\epsilon_s(t_j), \epsilon_v(t_j))$
9:         Compute bias $n_v(t_j)$ by counting $\{v_i \mid v_i \in (v_{t_j} - \epsilon_v(t_j)/2, v_{t_j} + \epsilon_v(t_j)/2)\}$
10:         Compute intrinsic reward $r_{t_j}^{\text{VCSE}} \leftarrow \psi(n_v(t_j) + 1)/d_s + \log \epsilon(t_j)$
11:         Compute total reward $r_{t_j}^{\text{T}} \leftarrow r_{t_j}^{\text{e}} + \beta \cdot r_{t_j}^{\text{VCSE}}$
12:     **end for**
13:     Update policy $\pi$, critic $f_{v,\text{T}}^\pi$ with $\{(s_{t_j}, a_{t_j}, s_{t_j+1}, r_{t_j}^{\text{T}})\}_{j=1}^B$
14:     Update extrinsic critic $f_v^\pi$ with $\{(s_{t_j}, a_{t_j}, s_{t_j+1}, r_{t_j}^{\text{e}})\}_{j=1}^B$
15: **end for**

---

corresponds to separately estimating the state entropies that are conditioned on the value estimates of each state and then maximizing their average. This intuitively can be seen as partitioning the visited state space with value estimates and averaging the state entropy of each partitioned space.

**Estimation and intrinsic reward**    To estimate $\mathcal{H}(S \mid V_\pi^{\text{vs}})$, we employ the KSG conditional entropy estimator in Equation 5. Specifically, we estimate the value-conditional state entropy as below:

$$\widehat{H}_{\text{KSG}}(S \mid V) = \frac{1}{N} \sum_{i=1}^N \Big[ \psi(n_v(i) + 1) + d_V(\log \epsilon(i) - \log \epsilon_v(i)) + d_S \log \epsilon(i) \Big] + C \qquad (6)$$

where $C$ denotes $-\psi(k) + \log c_{d_S}$. In practice, we maximize the lower bound of value-conditional state entropy because of its simplicity and ease of implementation as below:

$$\widehat{H}_{\text{KSG}}(S \mid V) \geq \frac{1}{N} \sum_{i=1}^N \Big[ \psi(n_v(i) + 1) + d_S \log \epsilon(i) \Big] - \psi(k) + \log c_{d_S} \qquad (7)$$

We then define the intrinsic reward $r_t^{\text{VCSE}}$ similarly to Equation 2 by ignoring constant terms as below:

$$r_t^{\text{VCSE}} = \frac{1}{d_S} \psi(n_v(i) + 1) + \log \epsilon(i) \text{ where } \epsilon(i) = 2 \cdot \max(||s_i - s_i^{k\text{NN}}||, ||v_i - v_i^{k\text{NN}}||) \qquad (8)$$

To provide an intuitive explanation of how our reward encourages exploration, we note that a $(s_j, v_j)$ pair whose value $v_j$ largely differs from $v_i$ is not likely to be chosen as $k$-th nearest neighbor because maximum-norm will be large due to large value norm $||v_i - v_j||$. This means that $k$-th nearest neighbor will be selected among the states with similar values, which corresponds to partitioning the state space with values. Then maximizing $\epsilon(i)$ can be seen as maximizing the state entropy within each partitioned space. We provide an illustration of our method in Figure 1.

## 4.2   Training

**Objective**    We train RL agents to solve the target tasks with the reward $r_t^{\text{T}} = r_t^{\text{e}} + \beta \cdot r_t^{\text{VCSE}}$ where $\beta > 0$ is a scale hyperparameter that adjusts the balance between exploration and exploitation. We do not utilize a decay schedule for $\beta$ unlike prior work (Seo et al., 2021) because our method is designed to avoid redundant exploration. We provide the pseudocode of our method in Algorithm 1.

**Implementation detail**    To compute the intrinsic reward based on value estimates of expected return from environments, we train an extrinsic critic $f_v^\pi$ to regress the expected return computed only with the extrinsic reward $r_t^{\text{e}}$. Then we use the value estimates from $f_v^\pi$ to compute the intrinsic reward in Equation 8. To train the agent to maximize $r_t^{\text{T}}$, we train a critic function $f_{v,\text{T}}^\pi$ that regresses

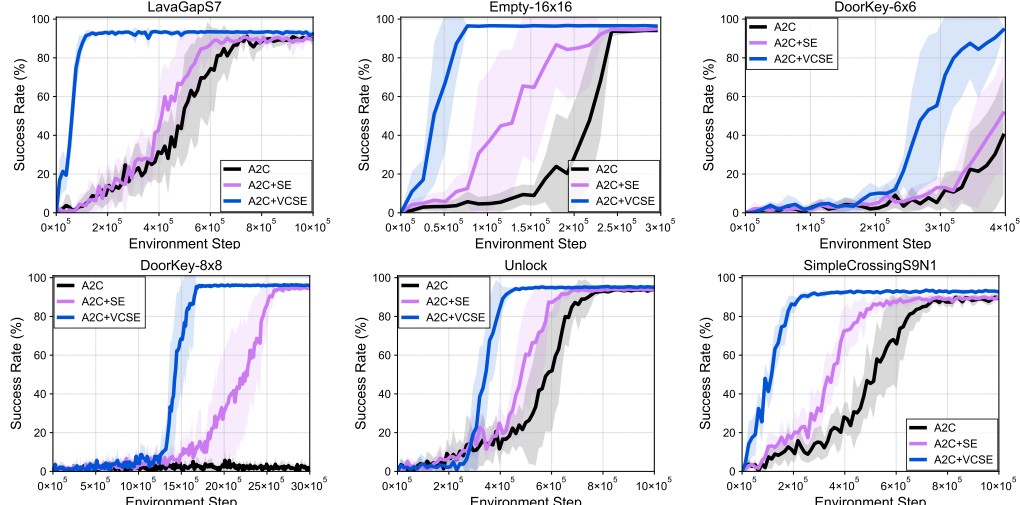

Figure 2: Learning curves on six navigation tasks from MiniGrid (Chevalier-Boisvert et al., 2018) as measured on the success rate. The solid line and shaded regions represent the interquartile mean and standard deviation, respectively, across 16 runs.

the expected return computed with $r_t^{\mathrm{T}}$, which the policy $\pi$ aims to maximize. To ensure that the distribution of value estimates be consistent throughout training, we normalize value estimates with their mean and standard deviation computed with samples within a mini-batch. When an environment is partially observable as a high-dimensional observation $o_t \in \mathcal{O}$ is only available instead of a fully observable state, we follow a common practice (Mnih et al., 2015) that reformulates the problem as MDP by stacking observations $\{o_t, o_{t-1}, o_{t-2}, ...\}$ to construct a state $s_t$. We also note that we use a replay buffer $\mathcal{B}$ for entropy estimation following Liu & Abbeel (2021b); Seo et al. (2021) in that it explicitly encourages the policy to visit unseen, high-reward states which are not in the buffer.

## 5 Experiments

We design our experiments to evaluate the generality of our maximum value-conditional state entropy (VCSE) exploration as a technique for improving the sample-efficiency of various RL algorithms (Mnih et al., 2016; Yarats et al., 2021a). We conduct extensive experiments on a range of challenging and high-dimensional domains, including partially-observable navigation tasks from MiniGrid (Chevalier-Boisvert

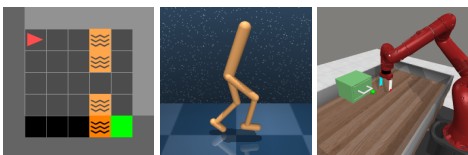

Figure 3: Examples of tasks from MiniGrid, DeepMind Control Suite, and Meta-World.

et al., 2018), pixel-based locomotion tasks from DeepMind Control Suite (DMC; Tassa et al. (2020)), and pixel-based manipulation tasks from Meta-World (Yu et al., 2020).

### 5.1 MiniGrid Experiments

**Setup** We evaluate our method on navigation tasks from MiniGrid benchmark (Chevalier-Boisvert et al., 2018) consisting of sparse reward goal-reaching tasks. This environment is partially observable as the agent has access to a $7 \times 7 \times 3$ encoding of the $7 \times 7$ grid in front of it instead of the full grid.[4] As a baseline, we first consider RE3 (Seo et al., 2021) which trains Advantage Actor-Critic (A2C; Mnih et al. 2016) agent with state entropy (SE) exploration where the intrinsic reward is obtained using the representations from a random encoder. We build our method upon the official implementation of RE3 by modifying the SE reward with our VCSE reward. Unlike RE3, we do not normalize our intrinsic reward with standard deviation and also do not utilize a separate buffer for computing the intrinsic reward using the on-policy batch. We use $k = 5$ for both SE and VCSE by following the original implementation. See Appendix A for more details.

---

[4]We provide additional results on fully-observable MiniGrid tasks in Appendix C, where trends are similar.

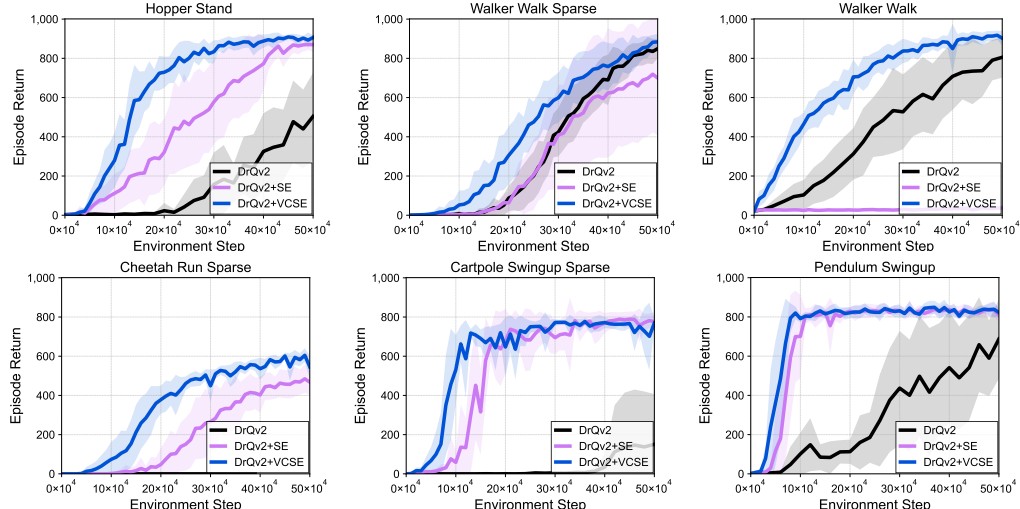

Figure 5: Learning curves on six control tasks from DeepMind Control Suite (Tassa et al., 2020) as measured on the episode return. The solid line and shaded regions represent the interquartile mean and standard deviation, respectively, across 16 runs.

**Results** Figure 2 shows that A2C+VCSE consistently outperforms A2C+SE on diverse types of tasks, including simple navigation without obstacles (Empty-16x16), navigation with obstacles (LavaGapS7 and SimpleCrossingS9N1), and long-horizon navigation (DoorKey-6x6, DoorKey-8x8, and Unlock). For instance, on LavaGapS7, A2C+VCSE achieves an average success rate of 88.8%, while A2C+SE achieves 13.9% after 100K steps. This shows that VCSE encourages the agent to effectively explore high-value states beyond a crossing point. On the other hand, SE excessively encourages the agent to visit states located before the crossing point for increasing the state entropy.

**Comparison to SE with varying** $\beta$ One might think that adjusting the scale of intrinsic rewards could address the issue of state entropy exploration in the supervised setup, by further encouraging exploration in high-value states or discouraging exploration in low-value states. However, as shown in Figure 4, simply adjusting the scale cannot address the issue. Specifically, A2C+SE fails to solve the task with a large $\beta = 0.05$, because large intrinsic rewards could make the agent ignore the task reward and encourages redundant exploration. On the other hand, small $\beta = 0.0005$ also degrades sample-efficiency when compared to the performance with $\beta = 0.005$ as it

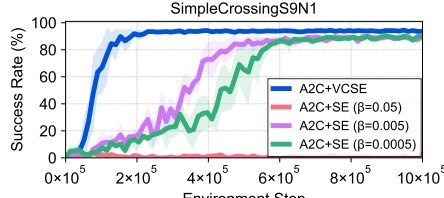

Figure 4: Learning curves on SimpleCrossingS9N1 as measured on the success rate. The solid line and shaded regions represent the interquartile mean and standard deviation, respectively, across eight runs.

discourages exploration. In contrast, A2C+VCSE learns to solve the task in a sample-efficient manner even with different $\beta$ values (see Figure 16 for supporting experiments).

### 5.2 DeepMind Control Suite Experiments

**Setup** We consider a widely-used DeepMind Control Suite (DMC; Tassa et al. 2020) benchmark mainly consisting of locomotion tasks. For evaluation on a more challenging setup, we conduct experiments on pixel-based DMC environments by using a state-of-the-art model-free RL method DrQv2 (Yarats et al., 2021a) as our underlying RL algorithm.[5] For computing the intrinsic bonus, we follow Laskin et al. (2021) by using the features from an intrinsic curiosity module (ICM; Pathak et al. 2017) trained upon the frozen DrQv2 encoder. For SE, we find that normalizing the intrinsic reward with its running mean improves performance, which is also done in Laskin et al. (2021). We do not normalize the intrinsic reward for our VCSE exploration. We also disable the noise scheduling

---

[5]We provide additional results on state-based DMC tasks in Appendix C, where trends are similar.

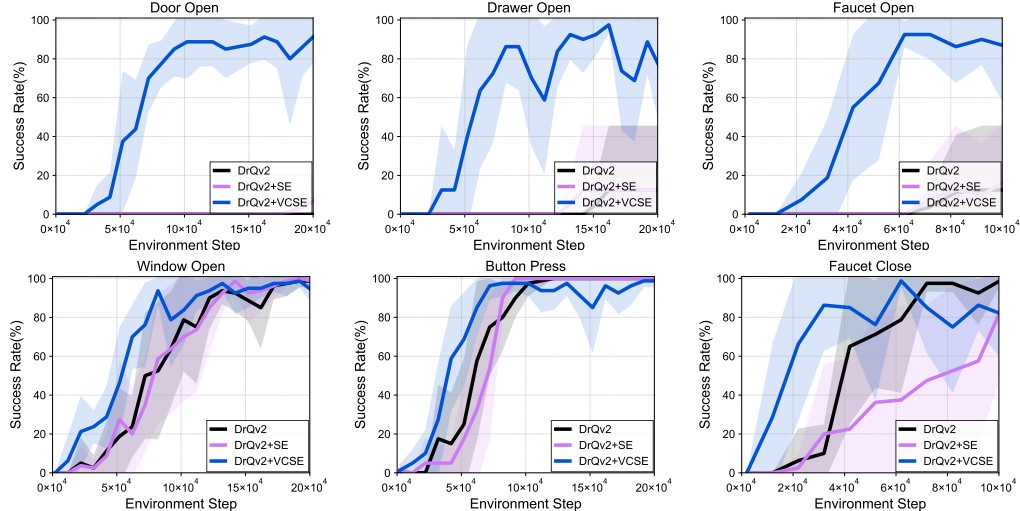

Figure 6: Learning curves on six visual manipulation tasks from Meta-World (Yu et al., 2020) as measured on the success rate. The solid line and shaded regions represent the interquartile mean and standard deviation, respectively, across 16 runs.

in DrQv2 for SE and VCSE as we find it conflicts with introducing the intrinsic reward. We instead use the fixed noise of $0.2$. We use $k = 12$ for both SE and VCSE. See Appendix A for more details.

**Results**   Figure 5 shows that VCSE exploration consistently improves the sample-efficiency of DrQv2 on both sparse reward and dense reward tasks, outperforming all other baselines. In particular, our method successfully accelerates training on dense reward Walker Walk task, while SE significantly degrades the performance. In Appendix B, we further show that the performance of DrQv2+SE improves with smaller $\beta$, but it still struggles to outperform DrQv2 on Walker Walk. This implies that SE might degrade the sample-efficiency of RL algorithms on dense reward tasks by encouraging the agent to explore states that might not be helpful for solving the task. Moreover, We show that introducing a decaying $\beta$ schedule for SE struggles to improve performance in Appendix B.

### 5.3   Meta-World Experiments

**Setup**   We further evaluate our method on visual manipulation tasks from Meta-World benchmark (Yu et al., 2020) that pose challenges for exploration techniques due to its large state space with small objects. For instance, moving a robot arm toward any direction enables the agent to visit novel states, but it would not help solve the target task. As an underlying RL algorithm, we use DrQv2 (Yarats et al., 2021a). We follow the setup of Seo et al. (2022a) for our experiments by using the same camera configuration and normalizing the extrinsic reward with its running mean to make its scale be 1 throughout training. For computing the intrinsic reward, we use the same scheme as in DMC experiments (see Section 5.2) by using ICM features for computing the intrinsic reward and only normalizing the SE intrinsic reward. We disable the noise scheduling for all methods and use $k = 12$ for SE and VCSE. See Appendix A for more details.

**Results**   Figure 6 shows that VCSE consistently improves the sample-efficiency of DrQv2 on visual manipulation tasks, while SE struggles due to the large state spaces. Notably, DrQv2+VCSE allows for the agent to solve the tasks where DrQv2 struggles to achieve meaningful success rates, *e.g.,* DrQv2+VCSE achieves 85% success rate on Door Open after 100K environment steps while DrQv2 achieves zero success rate. On the other hand, SE struggles to improve the sample-efficiency of DrQv2, even being harmful in several tasks. This shows the effectiveness of VCSE for accelerating training in manipulation tasks where state space is very large. We also report the performance of SE and VCSE applied to model-based algorithm (Seo et al., 2022a) on Meta-World in Appendix B, where the trend is similar in that SE often degrades the performance due to the large state space.

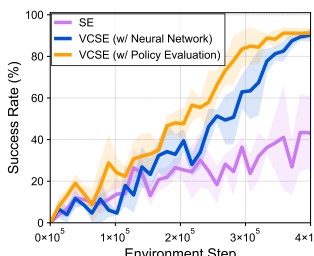 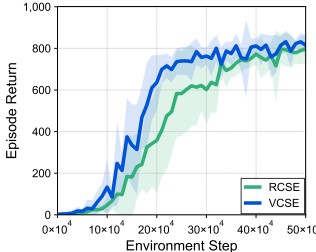 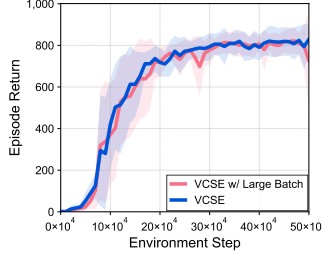

(a) Effect of value approximation  (b) Effect of value conditioning  (c) Effect of batch size

Figure 7: (a) Learning curves on SimpleCrossingS9N1 that compares VCSE using neural networks for value function approximation against VCSE that employs policy evaluation (Sutton & Barto, 2018) using the privileged simulator information. Learning curves aggregated on two control tasks from DeepMind Control Suite that investigate the effect of (b) value conditioning and (c) batch size. We provide results on individual task in Appendix B. The solid line and shaded regions represent the interquartile mean and standard deviation, respectively, across 16 runs.

## 5.4 Ablation Studies and Analysis

**Effect of value approximation**    Since we use neural networks for encoding states and approximating the value functions, our objective $\mathcal{H}(S|V_\pi^{\mathbf{vs}})$ could change throughout training. To investigate the effect of using the approximated value function for our objective, we provide experimental result where we (i) use ground-truth states for computing the intrinsic bonus and (ii) computes the value estimates via policy evaluation (Sutton & Barto, 2018) using the MiniGrid simulator. Figure 7a shows that VCSE based on policy evaluation using the privileged knowledge of ground-truth simulator performs the best, but VCSE based on value function approximation with neural networks can match the performance and significantly outperforms SE. This supports that our method works as we intended with the approximated value functions. We expect that our method can be further improved by leveraging advanced techniques for learning value functions (Bellemare et al., 2017; Espeholt et al., 2018; Hafner et al., 2023), reaching the performance based on the privileged knowledge.

**Effect of value conditioning**    We compare VCSE with a baseline that maximizes reward-conditional state entropy (RCSE) in Figure 7b to investigate our design choice of conditioning the state entropy on values instead of one-step rewards. The result shows that VCSE largely outperforms RCSE because the value captures more long-term information about the state when compared to the reward. We note that RCSE cannot be applied to navigation tasks in MiniGrid, where the reward is only given to goal-reaching states, which also demonstrates the benefit of VCSE in terms of applicability.

**Effect of batch size**    We investigate how estimating the value-conditional state entropy using samples from minibatches by reporting the results with increased batch sizes. Specifically, we use the batch size of 1024 for computing the intrinsic bonus but use the same batch size of 256 or 512 for training the actor and critic of RL agent. As shown in Figure 7c, we find that results are stable with both batch sizes, which shows that our value-conditional state entropy estimation can be stable without using extremely large batch sizes or all the samples within the replay buffer.

**Heatmap visualization**    To help understand how VCSE improves sample-efficiency, we provide the visualization of visited state distributions obtained during the training of A2C agents with SE and VCSE. For this analysis, we modified the original SimpleCrossingS9N1 task to have a fixed map configuration (see Appendix A for more details). As shown in Figure 8a, we find that the agent trained with SE keeps exploring the region before a narrow crossing point instead of reaching the goal until 100K steps, even if it has experienced several successful episodes at the initial phase of training. This is because the crossing point initially gets to have a high value and thus exploration is biased towards the wide low-value state region, making it difficult to solve the task by reaching the green goal. On the other hand, Figure 8b shows that VCSE enables the agent to successfully explore high-value states beyond the crossing point.

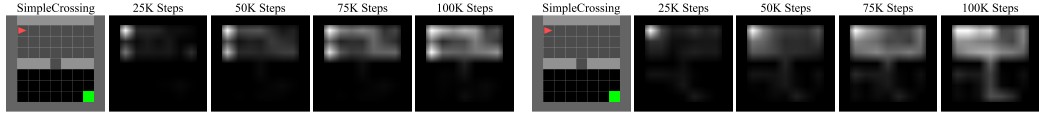

| (a) State entropy exploration | (b) Value-conditional state entropy exploration |

Figure 8: Visualization of visited state distribution during the training of A2C agent with (a) maximum state entropy (SE) exploration and (b) maximum value-conditional state entropy (VCSE) exploration to solve SimpleCrossing task from MiniGrid (Chevalier-Boisvert et al., 2018).

## 6 Discussion

**Limitation and future directions** One limitation of our work is that we lack a theoretical understanding of how value-conditional state entropy exploration works. Considering the strong empirical performance evidenced in our experiments, we hope our work facilitates future research on a theoretical investigation. Our work only considers a non-parametric state entropy estimator due to its benefit that does not learn a separate estimator, but considering learning-based estimator can be also an interesting direction. Moreover, applying our idea to methods that maximizes the state entropy of policy-induced state distributions (Lee et al., 2019; Mutti & Restelli, 2020) can be also an interesting future work. Our work might be limited in that it works exactly the same as the state entropy exploration until the agent encounters a supervised reward from an environment. Developing a method that utilizes an intrinsic signal to partition the state space into meaningful subspaces without the need for task reward is a future direction we are keen to explore in future work.

**Conclusion** We present a new exploration method that maximizes *value-conditional* state entropy which addresses the imbalanced distribution problem of state entropy exploration in a supervised setup by taking into account the value estimates of visited states for computing the intrinsic bonus. Our extensive experiments show that our method can consistently improve the sample-efficiency of RL algorithms in a variety of tasks from MiniGrid, DeepMind Control Suite, and Meta-World benchmarks. We hope our simple, intuitive, and stable exploration technique can serve as a standard tool for encouraging exploration in deep reinforcement learning.

## Acknowledgements and Disclosure of Funding

We want to thank anonymous reviewers and colleagues at KAIST ALIN-LAB and UC Berkeley RLL for providing helpful feedback and suggestions for improving our paper. We also thank Shi Zhuoran for helpful comments on the draft. This research is supported by Institute of Information & Communications Technology Planning & Evaluation (IITP) grant funded by the Korea government(MSIT) (No.2022-0-00953, Self-directed AI Agents with Problem-solving Capability; No.20190-00075, Artificial Intelligence Graduate School Program (KAIST)).

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

## Societal Impact

We do not anticipate immediate negative consequences from conducting this work because our experiments are based on simulation environments designed to conceptually evaluate the capabilities of reinforcement learning (RL) algorithms. Recent studies, however, demonstrate that large-scale RL when integrated with robotics can effectively work on real-world environments (Kalashnikov et al., 2021; Herzog et al., 2023). This makes it crucial to be aware of potential societal harms that RL agents could inadvertently cause. While this work does not aim to address these safety issues, our method might mitigate unintentional harm during the training process. For instance, it might prevent the agent from exhibiting potentially dangerous novelty-seeking behaviors, such as moving robot arms towards low-value, empty regions where a human researcher is likely to be situated.

## A   Experimental Details

**Compute**   For MiniGrid experiments, we use a single NVIDIA TITAN Xp GPU and 8 CPU cores for each training run. It takes 15 minutes for training the agent for 1M environment steps. For DeepMind Control Suite and Meta-World experiments, we use a single NVIDIA 2080Ti GPU and 8 CPU cores for each training run. It takes 36 minutes and 90 minutes for training the agent for 100K environment steps on DeepMind Control Suite and Meta-World benchmarks, respectively.

### A.1   Implementation Details

**Value normalization**   For normalizing value estimates to stabilize value-conditional state entropy estimation, we compute the mean and standard deviation using the samples within the mini-batch. We empirically find no significant difference to using the running estimate.

**Extrinsic critic function**   For training the extrinsic critic function described in Section 4.2, we introduce another set of critic and target critic functions based on the same hyperparameters used for the main critic the policy aims to maximize. Then we use the target critic for obtaining value estimates. We apply the stop gradient operation to inputs to disable the gradients from updating the extrinsic critic to update other components. For the policy, we use the same policy for training both main and extrinsic critic functions. We empirically find no need for training another policy solely for the extrinsic critic.

**A2C implementation details**   We use the official implementation[6] of RE3 (Seo et al., 2021) and use the same set of hyperparameters unless otherwise specified. Following the setup of RE3, we use a fixed, randomly initialized encoder to extract state representations and use them for computing the intrinsic reward. We use the same hyperparameter of fixed intrinsic scale $\beta = 0.005$ and $k = 5$ for both SE and VCSE following the original implementation. For RE3, we normalize the intrinsic reward with its standard deviation computed using the samples within the mini-batch, following the original implementation. But we do not normalize our VCSE intrinsic reward.

**DrQv2 implementation details**   We use the official implementation[7] of DrQv2 (Yarats et al., 2021a) and use the same set of hyperparameters unless otherwise specified. For both SE and VCSE exploration, we find that using $\beta = 0.1$ achieves the overall best performance. We also use $k = 12$ for both SE and VCSE. For computing the intrinsic reward, we follow the scheme of Laskin et al. (2021) that trains Intrinsic Curiosity Module (ICM; Pathak et al. 2017) upon the representations from a visual encoder and uses ICM features for measuring the distance between states. We note that we detach visual representations used for training ICM to isolate the effect of training additional modules on the evaluation. For both SE and VCSE exploration, we disable the noise scheduling scheme of DrQv2 that decays $\sigma$ from 1 by following a pre-defined schedule provided by the authors. This is because we find that such a noise scheduling conflicts with the approaches that introduce additional intrinsic rewards. Thus we use the fixed noise of $0.2$ for SE and VCSE exploration. For Meta-World experiments, we also disable the scheduling for the DrQv2 baseline as we find it performs better. But we use the original scheduling for the DrQv2 baseline following the official implementation.

---

[6]https://github.com/younggyoseo/RE3
[7]https://github.com/facebookresearch/drqv2

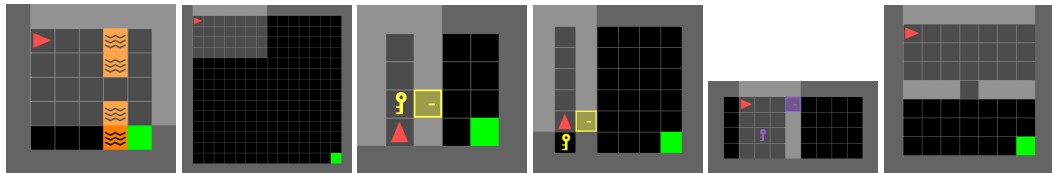

Figure 9: Examples of tasks in our MiniGrid experiments: (a) LavaGapS7, (b) Empty-16×16, (c) DoorKey-6×6, (d) DoorKey-8×8, (e) Unlock, and (f) SimpleCrossingS9N1.

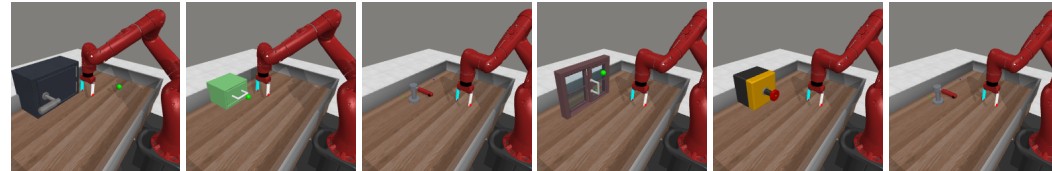

Figure 10: Examples of tasks we used in our Meta-World experiments: (a) Door Open, (b) Drawer Open, (c) Faucet Open, (d) Window Open, (e) Button Press, and (f) Faucet Close.

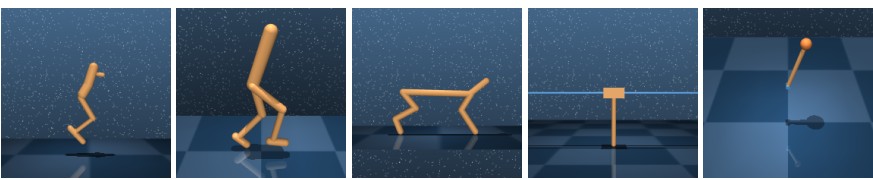

Figure 11: Examples of tasks we used in our DeepMind Control Suite experiments: (a) Hopper, (b) Walker, (c) Cheetah, (d) Cartpole, (e) Pendulum

**Heatmap analysis details**    For experiments in Figure 8, we use the easy version of SimpleCrossingS9N1 task from MiniGrid benchmark (Chevalier-Boisvert et al., 2018). Specifically, we disable the randomization of map configurations to make it possible to investigate the heatmap over a fixed map. For visualizing the heatmaps, we record $x, y$ position of agents during the initial 100K steps. We train A2C agent with both SE and VCSE exploration as we specified in Section 5.1 and Appendix A.1 without any specific modification for this experiment.

## A.2   Environment Details

**MiniGrid**    We conduct our experiments on six navigation tasks from MiniGrid benchmark (Chevalier-Boisvert et al., 2018): LavaGapS7, Empty-16×16, DoorKey-6×6, DoorKey-8×8, Unlock, and SimpleCrossingS9N1. We provide the visualization of the tasks in Figure 9. We use the original tasks without any modification for our experiments in Section 5.1.

**Meta-World**    We conduct our experiments on six manipulation tasks from Meta-World benchmark (Yu et al., 2020): Door Open, Drawer Open, Faucet Open, Window Open, Button Press, and Faucet Close. We provide the visualization of the tasks in Figure 10. We follow the setup of Seo et al. (2022a) that uses a fixed camera location for all tasks.

**DeepMind Control Suite**    We conduct our experiments on six locomotion tasks from DeepMind Control Suite benchmark (Tassa et al., 2020): Hopper Stand, Walker Walk Sparse, Walker Walk, Cheetah Run Sparse, Cartpole Swingup Sparse, and Pendulum Swingup. We use the sparse reward tasks introduced in Seyde et al. (2021), by following RE3. We provide the visualization of the tasks in Figure 10.

# B  Additional Experiments

## B.1  Experiments with Model-Based RL

**Setup**  As a model-based underlying RL algorithm, we consider Masked World Models (MWM; Seo et al. 2022a) that has shown to be able to solve more challenging, long-horizon tasks compared to DrQv2. We consider four tasks of various difficulties: Box Close, Handle Pull Side, Lever Pull, and Drawer Open. We use the official implementation[8] of MWM (Seo et al., 2022a) and use the same set of hyperparameters unless otherwise specified. For both SE and VCSE exploration, we find that using $\beta = 1$ performs best. Following the idea of Seo et al. (2022b) that introduces an additional reward predictor for intrinsic reward in the world model of DreamerV2 (Hafner et al., 2021), we introduce the reward network that predicts our intrinsic reward $r_t^{\text{VCSE}}$. For computing the intrinsic reward, we also follow the idea of Seo et al. (2022b) that uses a random projection (Bingham & Mannila, 2001) to reduce the compute cost of measuring distances between states. Specifically, we project 2048-dimensional model states into 256-dimensional vectors with random projection. Because the original MWM implementation normalizes the extrinsic reward by its running estimate of mean to make its scale 1 throughout training, we find that also normalizing intrinsic rewards with their running estimates of mean stabilizes training. We use $k = 12$ for both SE and VCSE.

**Results**  Figure 12 shows that VCSE consistently accelerates and stabilizes the training of MWM agents on four visual manipulation tasks of different horizons and difficulties, which shows that the effectiveness of our method is consistent across diverse types of RL algorithms. On the other hand, we observe that SE could degrade the performance, similar to our observation from experiments with DrQv2 on Meta-World tasks (see Figure 6). This supports our claim that SE often encourages exploration to be biased towards low-value states especially when high-value states are narrowly-distributed, considering that manipulation tasks have a very narrow task-relevant state space.

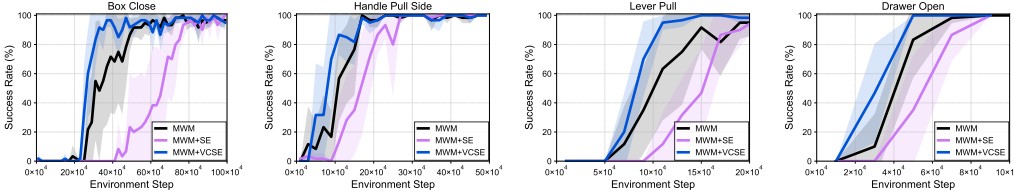

Figure 12: Learning curves on six visual manipulation tasks from Meta-World (Yu et al., 2020) as measured on the success rate. The solid line and shaded regions represent the interquartile mean and standard deviation, respectively, across 16 runs.

## B.2  Ablation Study

In Figure 13, we provide the results on individual task used for reporting the aggregate performance that investigate the effect of value conditioning and batch size (see Figure 7b and Figure 7c).

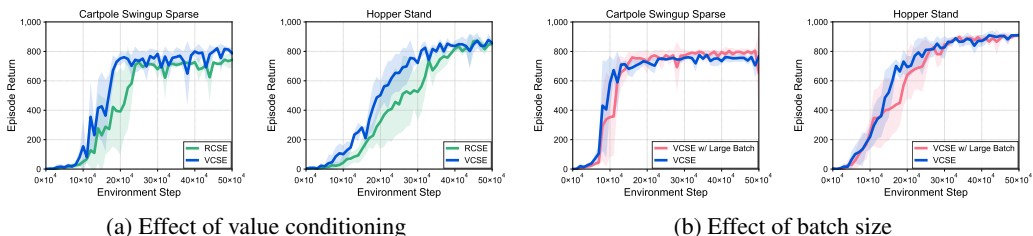

(a) Effect of value conditioning  (b) Effect of batch size

Figure 13: Learning curves on two visual locomotion control tasks from DeepMind Control Suite that investigate the effect of (a) value conditioning and (b) batch size. The solid line and shaded regions represent the interquartile mean and standard deviation, respectively, across eight runs.

---

[8]https://github.com/younggyoseo/MWM

## B.3  Experiments with Varying Intrinsic Reward Hyperparameters

**A2C+VCSE with varying $k$**    We provide additional experimental results with varying $k \in \{3, 5, 7\}$ on SimpleCrossingS9N1 task in Figure 14. We find that the performance of A2C+VCSE tends to degrade as $k$ increases. We hypothesize this is because higher $k$ leads to finding less similar states and this often leads to increased intrinsic reward scale.

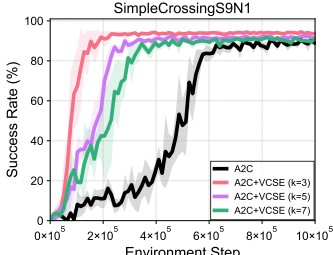

Figure 14: Learning curves as measured on the episode return. The solid line and shaded regions represent the interquartile mean and standard deviation, respectively, across eight runs.

**DrQv2+SE on walker walk with varying $\beta$**    We provide additional experimental results with varying $\beta \in \{0.1, 0.01, 0.001\}$ on Walker Walk task where DrQv2+SE significantly struggles to improve the sample-efficiency of DrQv2. In Figure 15, we find that the performance of DrQv2+SE is consistently worse than the vanilla DrQv2 with different $\beta$ values. This implies that adding SE intrinsic reward can be sometimes harmful for performance by making it difficult for the agent to exploit the task reward.

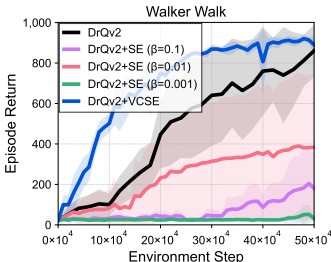

Figure 15: Learning curves as measured on the episode return. The solid line and shaded regions represent the interquartile mean and standard deviation, respectively, across eight runs.

**A2C+VCSE with varying $\beta$**    We also report the experimental results of A2C+VCSE with varying $\beta$ in Figure 16. We find that A2C+VCSE consistently improves A2C, in contrast to A2C+SE which fails to significantly outperform A2C even with different $\beta$ values (see Figure 4).

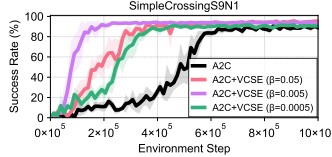

Figure 16: Learning curves as measured on the success rate. The solid line and shaded regions represent the interquartile mean and standard deviation, respectively, across eight runs.

**DrQv2+SE with decaying $\beta$**    We conduct additional experiments that compare DrQv2+VCSE with the state entropy baseline that uses decaying schedule for $\beta$, similarly to Seo et al. (2021) that uses $\beta$ schedule for the intrinsic reward. In Figure 17, we find that such a schedule cannot significantly

improve the performance of SE, except Hopper Stand where the performance is stabilized. Moreover, we find that the decaying schedule sometimes could degrade the performance, *i.e.,* Walker Walk Sparse. We also note that designing such a decaying schedule is a tedious process that requires researchers to tune the performance, making it less desirable even if it works reasonably well. We indeed find that the performance becomes very sensitive to the magnitude of decaying schedule.[9] On the other hand, DrQv2+VCSE exhibits consistent performance without the decaying schedule, which highlights the benefit of our approach that maximizes the value-conditional state entropy.

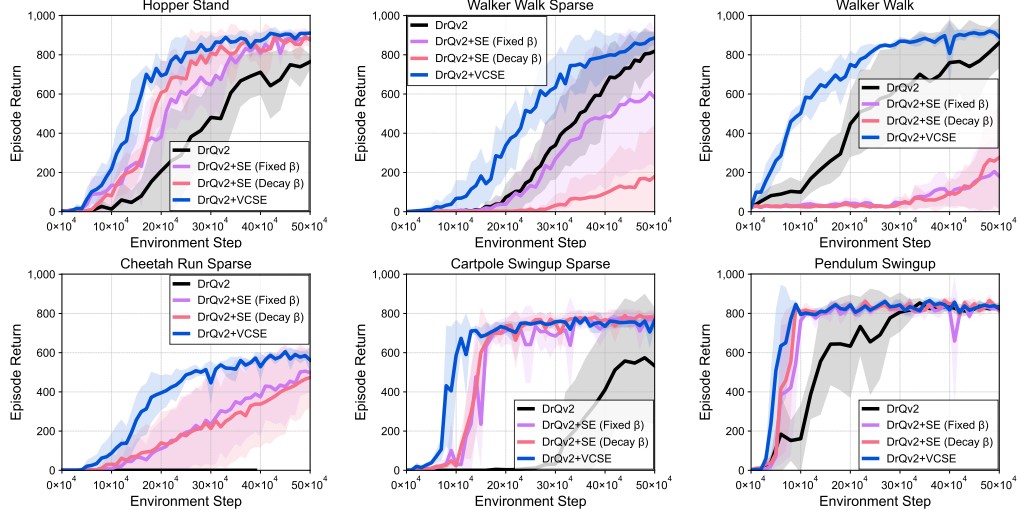

Figure 17: Learning curves on six visual locomotion control tasks from DeepMind Control Suite (Tassa et al., 2020) as measured on the episode return. The solid line and shaded regions represent the interquartile mean and standard deviation, respectively, across eight runs.

## C Experiments with Ground-Truth States

### C.1 MiniGrid Experiments

To demonstrate that our method also works in fully-observable MiniGrid where we do not use state encoder for computing the intrinsic bonus, we provide additional experiments that use fully observable states instead of partially observable grid encoding (see Section 5.1). Specifically, we use a set of one-hot vectors that represents a current map as inputs to the agent, and use the location of agent as inputs for computing the intrinsic bonus. Figure 18 shows that VCSE consistently accelerates the training, which highlights the applicability of VCSE to diverse domains with different input types.

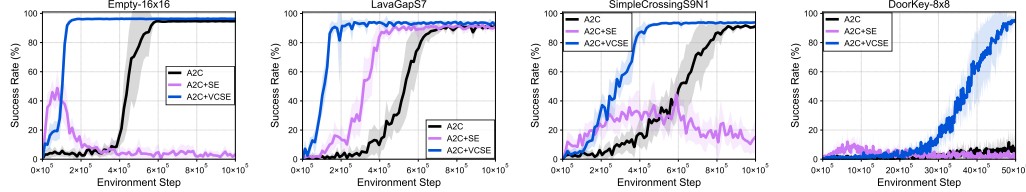

Figure 18: Learning curves on four navigation tasks from fully-observable MiniGrid (Chevalier-Boisvert et al., 2018) as measured on the success rate. The solid line and shaded regions represent the interquartile mean and standard deviation, respectively, across 16 runs.

---

[9]Due to the unstable performance from introducing the decay schedule, we run experiments with multiple decaying schedules and report the best performance for each task.

## C.2 DeepMind Control Suite Experiments

We further report experimental results in state-based DeepMind Control Suite experiments where we do not use state encoder for computing the intrinsic bonus. To make the scale of state norms be consistent across diverse tasks with different state dimensions, we divide the state input with its state dimension before computing the intrinsic bonus. As our underlying RL algorithm, we used Soft Actor-Critic (SAC; Haarnoja et al. 2018). For SE and VCSE, we disabled automatic tuning hyperparameter $\alpha$ and used lower value of $\alpha = 0.001$ in SAC, because we find that such an automatic tuning conflicts with introducing the intrinsic reward, similar to noise scheduling in DrQv2. Figure 19 shows that VCSE consistently accelerates the training, which highlights the applicability of VCSE to diverse domains with different input types.

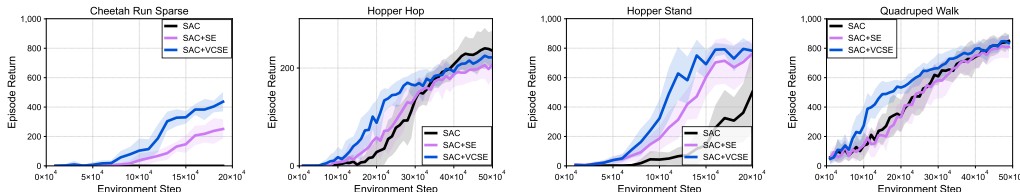

Figure 19: Learning curves on four locomotion tasks from state-based DeepMind Control Suite (Tassa et al., 2020) as measured on the success rate. The solid line and shaded regions represent the interquartile mean and standard deviation, respectively, across 16 runs.

## D   Additional Illustrations

We provide the additional illustration that helps understanding the procedure of estimating the conditional entropy with KSG estimator, which is explained in Section 3.2.

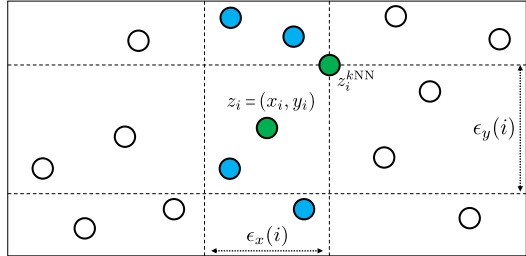

Figure 20: Illustration of a procedure for computing $\epsilon_x(i)$ and $n_x(i)$ when using the KSG estimator with $k = 2$. Given a centered point $z_i$, we first find $k$-nearest neighbor state $z_i^{k\mathrm{NN}}$. Then $\epsilon_x(i)$ is twice the distance from $x_i$ to $x_i^{k\mathrm{NN}}$ and $n_x$ can be computed as 5 by counting all the points located within $(x_i - \epsilon_x(i)/2, x_i + \epsilon_x(i)/2)$. We note that $\epsilon_y(i)$ and $n_y(i)$ can be also similarly computed.

