# OpenReview forum: "Accelerating Reinforcement Learning with Value-Conditional State Entropy Exploration"
_NeurIPS.cc/2023/Conference — NeurIPS 2023 poster_

### Official Review · Reviewer_FYPf · 2023-06-27

**Soundness:** 3 good
**Presentation:** 3 good
**Contribution:** 2 fair
**Rating:** 7
**Confidence:** 4

**Summary:**

This paper addresses the problem of sample-efficient exploration in sparse-rewards deep reinforcement learning. It builds on previous literature on the maximization of the state entropy as an exploration objective, which was mostly used in reward-free settings, proposing a value-conditioned state entropy objective along with the task reward. The latter pushes the agent to maximize the state entropy while penalizing the entropy over the estimated values of the visited states. This value-conditioned state entropy is implemented through an off-the-shelf non-parametric conditional entropy estimator. Finally, the paper compares the performance of value-conditioned entropy maximization against standard entropy maximization in MiniGrid, DeepMind Control, and Meta-World.

**Strengths:**

- The paper proposes an intrinsic reward can be easily incorporated in any previous method;
- The experiments show that the value-conditional objective brings improved/matching performance in a variety of domains;
- The paper clearly presents ideas and contribution, and also include some compelling visualization of the conditional entropy estimation.

**Weaknesses:**

- The intrinsic reward computation requires robust estimates of the value function, which is a challenge per se, especially in sparse-rewards domains;
- As mentioned by the authors in their conclusion, the theoretical ground for value-conditional state entropy is unclear.

**Questions:**

This paper advocates for using state entropy bonuses to accelerate RL in sparse-rewards tasks, instead of training an entropy maximizing policy in reward-free settings, which is the most common mode of previous works.
To adapt state entropy bonuses to the demand of the sparse-reward task, which requires to deal with the exploration-exploitation trade-off, the paper proposes to condition the entropy of the visited states with the value of those states, effectively driving exploration towards rewarding regions of the state space.
This interesting notion leads to a methodology that can be easily incorporated in previous algorithms while benefitting the resulting sample efficiency in a variety of domains.
However, the paper does not provide a formal theoretical understanding of the proposed approach, which leaves some doubts on the generality of their results.
Anyway, I am currently providing a slightly positive score while encouraging the authors to improve the theoretical ground for value-conditioned entropy bonuses, and especially how the estimation error of the value function impacts the learning process.

**(Clarification on the objective)**

From my understanding of Algorithm 1, the proposed method maximizes the value-conditional state entropy over the samples taken from the replay buffer, and not just the samples drawn with the latter policy. While this is common in previous works as well (e.g., Liu and Abbeel, 2021), I am wondering whether looking at the value-conditional entropy of the latest samples is more appropriate in a setting where external reward is also present. Can the authors discuss their implementation choice and comment pros and cons of the two alternatives?

**(Sensitivity to $\beta$ also for VCSE)**

The experiments report an interesting analysis on how the value of $\beta$ affects the performance of A2C+SE. It would be interesting to inspect also the sensitivity of VCSE to the corresponding $\beta$ value.

**(Comparison with CEM)**

A recent work (Yang and Spaan, CEM: Constrained entropy maximization for task-agnostic safe exploration, 2023) considers state entropy maximization under cost constraints. I am wondering whether their approach could be also used to address the problem of reducing task-irrelevant exploration, e.g., by introducing appropriate value constraints. Can the authors discuss the comparison between their solution and CEM, possibly highlighting pros and cons of the two?

**Limitations:**

Some limitations of the paper and the proposed methodology are reported in the first paragraph of Section 6. I believe that a formal discussion on how the value estimation error affects their method would be extremely valuable.

---

> ### Author Rebuttal · Authors · 2023-08-08
>
>
> Dear Reviewer FYPf,
>
> We sincerely appreciate your efforts and insightful comments to improve the draft. We respond to each of your comments one-by-one in what follows.
>
> ---
>
> **Q1. Importance of value estimation**
>
> **A1.** In Figure 7(a) of the original draft, we reported that the performance further improves when using the ground-truth value estimated with privilege information from the simulator. As you mentioned, this implies that the performance of our method can indeed depend on the quality of learned value functions. Further quantitative evaluation on the relation between the quality of value functions and the performance of our method is indeed an interesting question, but we leave it to future work as measuring the quality of value functions is an open problem. It would be also very appreciated if the reviewers might perhaps be able to suggest experimental designs for this analysis experiment.
>
> --------
>
>
> **Q2. Discussion on the pros and cons of maximizing value-conditional state entropy using the samples from a replay buffer**
>
> **A2.** This is an interesting question. *The benefit of maximizing the entropy using the samples from a replay buffer* is that it explicitly encourages the policy to visit unseen states which are not in a replay buffer yet. This is more aligned with a supervised RL setup where the goal of exploration is supporting the policy to find a novel state with a high extrinsic reward. We adopted this buffer-based strategy in our work because our method is designed for the supervised RL setup. On the other hand, *maximizing the entropy of a policy-induced state distribution* is beneficial in case we want to learn a policy that can visit diverse states. For instance, this can be particularly useful in an unsupervised RL setup where the goal is to train a policy that can be quickly adapted to solve the downstream task when the reward becomes available. We will include the relevant discussion in the final draft.
>
> --------
>
> **Q3. Analysis on the effect of $\beta$.**
>
> **A3.** Following your suggestion, we conducted additional experiments with varying $\beta$ for VCSE. As shown in Figure 2 available from the attached PDF in the general response, we find that A2C+VCSE consistently improves A2C, in contrast to A2C+SE which fails to significantly outperform A2C even with different $\beta$ values.
>
> --------
>
> **Q4. comparison with CEM**
>
> **A4.** Thanks for your suggestion to include additional discussion on relevant work [Yang and Spaan, 2023]. As you mentioned, one can think of an idea to discourage the exploration in task-irrelevant state space by introducing a value constraint similar to CEM that introduces a safety constraint. This idea can be also useful in our setup in that it can introduce a scheduling for value constraint so that it can first encourage the agent to visit all the states and thus focus on exploration around high-value regions.
>
> However, we would like to note that VCSE is based on a different motivation in that VCSE aims to prevent the states with different high- (or low-) values from affecting exploration around low- (or high-) value states, not to discourage exploration on specific state regions.  Moreover, we note that a mechanism to similar value constraint scheduling can be observed when the environment is sparsely-rewarded because VCSE acts the same as SE until it observes a sparse reward. We will include the relevant discussion in the final draft. We will include the relevant discussion in the final draft.
>
> [Yang and Spaan., 2023] Yang, Qisong, and Matthijs TJ Spaan. "CEM: Constrained Entropy Maximization for Task-Agnostic Safe Exploration." The Thirty-Seventh AAAI Conference on Artificial Intelligence. 2023.

---

> > ### Comment · Reviewer_FYPf · 2023-08-18
> > **After response**
> >
> > I want to thank the authors for their detailed response, which is properly addressing my concerns. I am updating my evaluation upwards, and I really encourage the authors to include some bits of the provided clarifications and additional sensitivity analysis to a final draft of the paper.
> >
> > As for how to evaluate the impact of approximate value functions. I think the best way to do so would be through a theoretical analysis that shows how the value function error propagates to the objective function. Of course, this might be non-trivial. In terms of experiments, it would be great to compare the performance of VCSE with a learned value function against "fixed" value approximators, which I guess would derail learning completely when the quality of the approximation is not good enough. Also, an ablation study on timescale separation between value learning and policy learning would be interesting.

---

> > > ### Author Response · Authors · 2023-08-21
> > >
> > > Dear reviewer FYPf,
> > >
> > > Thank you for your reply! We will make sure that our clarifications and additional experiments are fully incorporated into the final draft. Moreover, we would really appreciate your suggestions on additional experiments with fixed value functions and separated actor-critic learning phases. We will try our best to design a more concrete experimental setup and include the results in the final draft.
> > >
> > > Thank you very much,
> > >
> > > Authors

---

### Official Review · Reviewer_pL2M · 2023-07-05

**Soundness:** 3 good
**Presentation:** 3 good
**Contribution:** 3 good
**Rating:** 6
**Confidence:** 4

**Summary:**

This paper present a exploration technique that maximizes the value-conditional state entropy, which separately estimates the state entropies that are conditioned on the value estimates of each state, then maximizes their average. By only considering the visited states with similar value estimates for computing the intrinsic bonus, it prevents the distribution of low-value states from affecting exploration around high-value states, and vice versa. The experiments demonstrate that the proposed alternative to the state entropy baseline significantly accelerates various reinforcement learning algorithms across a variety of tasks within MiniGrid, DeepMind Control Suite, and Meta-World benchmarks.


**Strengths:**

(1) The exploration method proposed in this paper show that maximum value-conditional state entropy (VCSE) exploration successfully  accelerates the training of RL algorithms. It can be used for reference in other RL algorithms.

(2) For section 3, the development process of the entropy estimator is very clearly described.

(3) For section 5, extensive experiments have been conducted to demonstrate the effectiveness of the algorithm proposed in this paper.


**Weaknesses:**

(1) The writing of this article seems difficult to understand in some aspects，especially for certain formulas.

(2) The motivation of this article is not fully explained.

(3) To be honest，it lacks a certain degree of novelty. As is well known, the methods used in this paper like “k-nearest neighbor entropy estimator” and “KSG conditional entropy estimator” are all previous work. We just utilize some of them.

(4) The main body of this paper is clear to understand, here is space for improvement. I defer some of my issues in the appendix to "Questions".


**Questions:**

(1) In line 117, what is the meaning of  z and z’?

(2) In Equ.(1)，what is the meaning of dx ? It seems unexplained.

(3) In section 5, the selected experimental environment is relatively simple with a low dimension. Can we select some complex environments with a high-dimension state and action spaces for verification?

(4) In Fig1, how to do “State Norm” and “Value Norm”?

(5) In Fig1, why do you only choose the third one(kNN (k=3))instead of the others？


**Limitations:**

In section 6, the authors talk about some limitation and future directions about this paper. At present, the potential negative social impact of this work has not been identified.

---

> ### Author Rebuttal · Authors · 2023-08-08
>
> Dear Reviewer pL2M,
>
> We sincerely appreciate your efforts and insightful comments to improve the draft. We respond to each of your comments one-by-one in what follows.
>
> ---
>
> **Q1. Motivation and Novelty**
>
> **A1.** In this work, we aim to improve the sample-efficiency of deep RL algorithms by introducing a novel exploration technique that can address the problem of balancing exploration and exploitation, which is an important and challenging problem for deep RL as highlighted by Reviewer MwTR and FYPf. Specifically, we propose a new exploration technique that builds upon a widely-used state entropy (SE) exploration by pointing out that SE suffers from an imbalance between the distributions of high-value and low-value states and introducing a new objective that takes into account the value estimates of states.
>
> We would also like to emphasize that our key novelty lies in introducing a new objective that maximizes the value-conditional state entropy $H(S|V)$ instead of the state entropy $H(S)$. To the best of our knowledge, our work is first to propose this objective and demonstrate its effectiveness. Moreover, we would like to note that which estimator to use is one of our design choices and not the main focus of this work. As Reviewer MwTR mentioned in their review, any entropy/mutual information estimator is compatible with our method. We will try to further clarify our motivation and novelty in the final draft.
>
> --------
>
>
> **Q2. What is the meaning of $z$ and $z’$ in Line 117?**
>
> **A2.** $z$ and $z’$ are arbitrary two samples from a random variable Z = (X, Y), which is a notation used for introducing the concept of the maximum norm. Thanks for pointing this out and we will clarify this in the final draft.
>
> --------
>
> **Q3. The meaning of $d_{X}$ in Equation 1.**
>
> **A3.** $d_{X}$ is the dimensionality of a random variable $X$. Similarly, $d_{S}$ and $d_{Y}$ are the dimensionalities of random variables $S$ and $Y$, respectively. Thanks for pointing this out and we will fix this in the final draft.
>
> --------
>
> **Q4. Additional experiments on more complex environments**
>
> **A4.** We would like to emphasize that our experiments are conducted in environments where the observation and action spaces are high-dimensional. For instance, we reported main experimental results on pixel-based environments where the inputs to the agents are high-dimensional multi-channel inputs. Moreover, we note that our method is effective on Meta-World environments where the robot arm can freely move inside the wide robot workspace so that state-action space becomes very high-dimensional.
>
> Nonetheless, to further address your concern, we provide additional experimental results on the Quadruped Walk task which has more high-dimensional state/action spaces compared to other considered tasks like Hopper. As shown in Figure 1 available from the attached PDF in the general response, we find that our VCSE clearly outperforms SE especially in the initial phase of training, which implies that the effectiveness of our method is consistent on even more complex tasks. We will include relevant results in the final draft.
>
> --------
>
> **Q5.  How to compute state and value norms in Figure 1?**
>
> **A5.** To compute the state norm in Figure 1, we randomly sample states from a replay buffer and compute the Euclidean norm between states. For computing the value norm, we pass these states through the critic function to get value estimates, and compute the Euclidean norm between value estimates. We will add a more detailed explanation on the caption for Figure 1 in the final draft.
>
> --------
>
> **Q6.  In Figure 1, why do you only choose the third one(kNN (k=3))instead of the others?**
>
> **A6.** We note that $k$ is a hyperparameter in our method and basically we can use any $k$ for our method. We choose $k=3$ in Figure 1 for illustrative purposes. We will clarify this in the final draft.

---

> > ### Comment · Reviewer_pL2M · 2023-08-18
> >
> > In this round of feedback, the author provided detailed explanations and modifications to the questions I raised. Especially the description of novelty and innovation can enhance the overall understanding of the article, and it is recommended to supplement it in the relevant parts of the article. In terms of expanding the experiment, in order to dispel my doubts, the author added relevant experiments to support it, which makes the article more competitive. Finally, I suggest that the author further sort out the entire article to make its context clear. Therefore, I agree to increase the article by 1 points.

---

> > > ### Author Response · Authors · 2023-08-21
> > >
> > > Dear reviewer pL2M,
> > >
> > > Thank you for your reply! We'll make sure that our additional clarification, experiments, and editorial comments are fully incorporated into the final draft.
> > >
> > > Thank you very much,
> > >
> > > Authors

---

### Official Review · Reviewer_MwTR · 2023-07-07

**Soundness:** 4 excellent
**Presentation:** 4 excellent
**Contribution:** 4 excellent
**Rating:** 7
**Confidence:** 4

**Summary:**

The paper proposes an improvement over a popular intrinsic reward based on state entropy. Instead of encouraging large state entropy uniformly over all states which may deviate the policy toward failure states, the proposed intrinsic reward motivates the agent to maximize the value-conditioned state entropy. The value-conditioned state entropy is estimated with a classical kNN method.

**Strengths:**

The intrinsic reward is an important direction of improving RL exploration and sample efficiency, and how to balance between instrinsic and extrinsic reward is a challenge for intrinsic reward design. This paper proposes a novel method to achieve this goal --- conditioning the state entropy on the state value to partition the state space and focus on state entropy over states with similar value. The paper is well written and the proposed method is strongly supported by a large range of experiments.

**Weaknesses:**

There is no significant weakness that I notice except for an ablation study on the effect of k for the kNN.

**Questions:**

Besides the effect of k, I wonder how if authors have tried other entropy/mutual information estimator (for example, the one based on contrastive learning) and why ending up selecting the kNN one.

**Limitations:**

kNN method for entropy estimation usually suffers in high-dimensional space, but the proposed method works well in environments with visual observations. So I don't find any significantly limitation.

---

> ### Author Rebuttal · Authors · 2023-08-08
>
>
> Dear Reviewer MwTR,
>
> We sincerely appreciate your efforts and insightful comments to improve the draft. We respond to each of your comments one-by-one in what follows.
>
> ---
>
> **Q1. Analysis on the effect of $k$**
>
> **A1.** Following your suggestion, we conducted additional experiments with $k \in \{3, 5, 7\}$ and observed that the performance of our method tends to degrade as $k$ increases As shown in Figure 3 available from the attached PDF in the general response, We hypothesize this is because higher $k$ leads to finding less similar states and this often leads to increased intrinsic reward scale. We will include the relevant results in the final draft.
>
> ---
>
> **Q2. Other entropy/mutual information estimator?**
>
> **A2.** It’s an interesting question. We employed a $k$-NN estimator following prior work, mainly because it’s non-parametric so that it does not incur additional training cost of training another neural network based estimator. But investigating the possibility of leveraging other estimators for online RL is definitely an interesting future direction and we will include the relevant discussion in the final draft. Thank you!

---

> > ### Comment · Reviewer_MwTR · 2023-08-13
> >
> > Thanks for your responses and they answer my questions clearly!

---

> > > ### Author Response · Authors · 2023-08-14
> > >
> > > Dear reviewer MwTR,
> > >
> > > Thank you for your reply! We're excited to hear that our response successfully answered your questions. We'll make sure that our rebuttal response is fully incorporated into the final draft.
> > >
> > > Thank you very much,
> > >
> > > Authors

---

### Official Review · Reviewer_CsYB · 2023-07-17

**Soundness:** 2 fair
**Presentation:** 3 good
**Contribution:** 2 fair
**Rating:** 4
**Confidence:** 3

**Summary:**

The paper investigates an exploration technique that maximizes the entropy of visited states while taking into account the expected return of the states. The goal is to explore the part of the state space that is both less visited while avoiding too much exploration for the low-value states.

**Strengths:**

The paper is overall well written and presents an interesting approach to taking into account an estimate of the expected return together with state entropy exploration.

**Weaknesses:**

My main concern is that it is a bit unclear how the presented algorithm actually enforces more exploration on the high expected return states as compared to previous approaches. In Figure 8, I'm unsure how VCSE will have a different behaviour than SE given that the agent needs to at least visit once the reward to be able to see that there is a lower bonus by keeping exploring the low expected return states. In addition, previous approaches that use a combination of intrinsic and extrinsic rewards should also tend to visit more the high expected return parts of the state space.

Additional minor comments:
- I'm unsure why "supervised setup" is mentioned in line 4 (abstract) and line 29.
- line 91: $\gamma$ can't be 1.
- Typos: line 105

**Questions:**

Can you clarify my main concern (see above). If so, I would happily increase my score.

**Limitations:**

The authors adequately address limitations.

---

> ### Author Rebuttal · Authors · 2023-08-08
>
> Dear Reviewer CsYB,
>
> We sincerely appreciate your efforts and insightful comments to improve the draft. We respond to each of your comments one-by-one in what follows.
>
> --------
>
> **Q1. Unclear how the presented algorithm actually enforce more exploration on the high expected return state as compared to previous approach.**
>
> **A1.** We would like to clarify that VCSE does not aim to encourage more exploration only on high-value state space, but aims to prevent a scenario where the exploration is biased towards specific regions within the state space. Specifically, VCSE prevents the states with drastically different value estimates from having an effect on computing the intrinsic reward. This prevents the distribution of low- (or high-) value states from affecting exploration around high- (or low-) value states. Thus VCSE can encourage uniform exploration within each value-conditional state space, which is the main difference to the previous approaches that uses a combination of intrinsic and extrinsic reward for exploration. We will try to further clarify this in the final draft.
>
> For instance, in Figure 8 of the original draft, one can expect that VCSE and SE would behave exactly the same before it encounters a reward from an environment. We observed that both methods indeed experience successful episodes and the agent learns to visit more states within the optimal path towards the goal at similar steps for both methods. This starts to make a visible difference between SE and VCSE. Specifically, this high visitation count within the optimal path leads to higher SE intrinsic rewards for non-optimal states outside the path, which biases exploration towards non-optimal states. On the other hand, VCSE encourages exploration on both state regions within and outside the optimal path, allowing for quickly learning to solve the target tasks.
>
> --------
>
> **Q2. I'm unsure why "supervised setup" is mentioned in line 4 (abstract) and line 29.**
>
> **A2.** We use the “supervised setup” term to denote that we consider a setup where the external task reward is available from the environment, following the terminology used in [Laskin et al., 2021].
>
> [Laskin et al., 2021] Laskin, Michael, Denis Yarats, Hao Liu, Kimin Lee, Albert Zhan, Kevin Lu, Catherine Cang, Lerrel Pinto, and Pieter Abbeel. "URLB: Unsupervised reinforcement learning benchmark." arXiv preprint arXiv:2110.15191 (2021).
>
> --------
>
> **Q3. $\gamma \in [0, 1]$**
>
> **A3.** We followed the notation of [Sutton & Barto, 2018] where the discount factor $\gamma$ is defined to be located within $[0, 1]$. We also note that $\gamma$ can be set to 1 if the task is episodic so that there is a terminating state [Pitis, 2019]
>
> [Sutton & Barto, 2018] Sutton, Richard S., and Andrew G. Barto. Reinforcement learning: An introduction. MIT press, 2018.
>
> [Pitis, 2019] Pitis, Silviu. "Rethinking the discount factor in reinforcement learning: A decision theoretic approach." In Proceedings of the AAAI Conference on Artificial Intelligence, vol. 33, no. 01, pp. 7949-7956. 2019.
>
> --------
>
> **Q4. Typos error in line 105**
>
> **A4** Thank you for pointing this out. We’ll fix this in the final draft.

---

> > ### Comment · Reviewer_CsYB · 2023-08-16
> > **The answer didn't clarify my doubts**
> >
> > In Fig 8, the sub figures a) and b) seem to have only one reward at the bottom right. It is still unclear to me why VCSE works better in this case.
> >
> > For the minor comment, $\gamma$ can't be one if the horizon is infinite (which seems to be the case here). The expected return would be unbounded.

---

> > > ### Author Response · Authors · 2023-08-17
> > > **Further Clarification by Authors**
> > >
> > > Dear Reviewer CsYB, we respond to each of your comments one-by-one in what follows.
> > >
> > > A1. Further explanation on how VCSE can work better than SE in Figure 8.
> > >
> > > First of all, we would like to clarify two things. In Figure 8, RL agents should still do additional exploration for exploiting the task reward after encountering the first reward, because the observation is partially observable and thus there could be states with higher value estimates. Moreover, both SE and VCSE obtain rewards at the similar timestep while it is not that visible in the heatmap from Figure 8.
> > >
> > > Now we will provide a more detailed explanation of how VCSE can work better than SE in this setup.  Because the RL agent tries to exploit the task reward, the agent initially begins to visit more states around the high-value states. However, because SE aims to encourage uniform coverage by visiting both low-value and high-value states similarly, the intrinsic reward for low-value states begins to increase. Then the agent starts to do excessive exploration around low-value states, instead of further exploring states around the rewarding state at the bottom right state in the map. This makes it difficult for the A2C+SE agent to quickly learn to solve the target task. Moreover, we further showed that this issue is not easily addressed by adjusting the scale of intrinsic rewards in Figure 4.
> > >
> > > On the other hand, VCSE can avoid this issue because it does not consider states with different values when calculating intrinsic rewards. This means that VCSE can still encourage exploration around high-value states, which allows the agent to visit states with higher values and quickly learns to solve the target task.
> > > We hope that this explanation answers your question. Please let us know if there still is anything not clear, we will try to further clarify this in the final draft.
> > >
> > > —
> > >
> > > A2. Discount factor
> > >
> > > Thank you for pointing this out. We will incorporate your comment in the final draft.

---

### Author Rebuttal · Authors · 2023-08-08

Dear reviewers,

We sincerely appreciate your efforts and insightful comments. Following your suggestions, we provide a one-page pdf that contains (i) the analysis on the effect of $\beta$ and $k$ and (ii) additional results on a high-dimensional Quadruped environment. We will incorporate them into the final draft.

Sincerely,

Authors

---

### Decision · Program_Chairs · 2023-09-21

**Decision:**

Accept (poster)

**Comment:**

This paper has been positively evaluated by the majority of reviewers. Only one reviewer expressed concerns about the significance of the proposed work compared to other approaches/methods in the literature. These concerns have been extensively addressed by the authors in the rebuttal. Even if the reviewer has not been fully convinced and kept the grade of borderline reject. Considering the outcome of the discussion phase, it is clear that the average opinion of the reviewers is positive and that the remaining concerns are not critical.

I encourage the authors to address all the comments and to incorporate the recommended improvements in the final version.